# Visible-Light-Driven Degradation of Biological Contaminants on the Surface of Textile Fabric Modified with TiO_2_-N Photocatalyst

**DOI:** 10.3390/ijms26157550

**Published:** 2025-08-05

**Authors:** Maria Solovyeva, Evgenii Zhuravlev, Yuliya Kozlova, Alevtina Bardasheva, Vera Morozova, Grigory Stepanov, Denis Kozlov, Mikhail Lyulyukin, Dmitry Selishchev

**Affiliations:** 1Research and Educational Center “Institute of Chemical Technologies”, Novosibirsk State University, Pirogova St. 2, Novosibirsk 630090, Russia; m.soloveva9@g.nsu.ru (M.S.); d.kozlov@g.nsu.ru (D.K.); m.lyulyukin@g.nsu.ru (M.L.); 2Competence Center of the National Technological Initiative “Hydrogen as the Basis of a Low-Carbon Economy”, Lavrentieva Ave. 7, Novosibirsk 630090, Russia; 3Institute of Chemical Biology and Fundamental Medicine SB RAS, Lavrentieva Ave. 8, Novosibirsk 630090, Russia; evgenijur@gmail.com (E.Z.); stepanovga@niboch.nsc.ru (G.S.)

**Keywords:** self-cleaning textile, photoactive fabric, photocatalytic degradation, N-doped TiO_2_, visible light

## Abstract

The problem of spreading harmful infections through contaminated surfaces has become more acute during the recent coronavirus pandemic. The design of self-cleaning materials, which can continuously decompose biological contaminants, is an urgent task for environmental protection and human health care. In this study, the surface of blended cotton/polyester fabric was functionalized with N-doped TiO_2_ (TiO_2_-N) nanoparticles using titanium(IV) isopropoxide as a binder to form durable photoactive coating and additionally decorated with Cu species to promote its self-cleaning properties. The photocatalytic ability of the material with photoactive coating was investigated in oxidation of acetone vapor, degradation of deoxyribonucleic acid (DNA) fragments of various lengths, and inactivation of PA136 bacteriophage virus and *Candida albicans* fungi under visible light and ultraviolet A (UVA) radiation. The kinetic aspects of inactivation and degradation processes were studied using the methods of infrared (IR) spectroscopy, polymerase chain reaction (PCR), double-layer plaque assay, and ten-fold dilution. The results of experiments showed that the textile fabric modified with TiO_2_-N photocatalyst exhibited photoinduced self-cleaning properties and provided efficient degradation of all studied contaminants under exposure to both UVA and visible light. Additional modification of the material with Cu species substantially improved its self-cleaning properties, even in the absence of light.

## 1. Introduction

The growing density of urban populations accelerates the spread of infectious diseases [1]. Research on pathogen transmission and disease control methods remains a critical focus for the scientific community. The COVID-19 pandemic highlighted the risks of surface-mediated infection transmission, underscoring the urgent need for materials capable of permanent surface decontamination [2,3]. Self-cleaning paints [4,5], window glasses [6,7], tiles, and filters [8] have been developed. Among these, self-cleaning textiles warrant special attention, as fabrics can harbor pathogenic microorganisms that remain infectious for long periods [9]. Antibacterial properties can be achieved via the decoration of a fabric’s surface with silver and copper nanoparticles [10,11]. The functionalization of fabrics with photocatalysts, such as titanium dioxide (TiO_2_) and zinc oxide (ZnO), is another promising approach to obtain self-cleaning textiles with a high oxidation activity in chemical degradation and microorganism disinfection [12]. TiO_2_-coated fabrics can completely decompose chemical pollutants [13,14,15] and provide inactivation of biological agents under exposure to light [16,17].

As TiO_2_ has a bandgap of 3.0–3.2 eV [18], the TiO_2_-coated fabrics can be activated under UV radiation only. ZnO is also commonly used as the photoactive component for the modification of textile materials to provide them with self-cleaning properties. Similarly to TiO_2_, the ZnO-coated fabrics can decompose contaminants under the radiation of UV region due to the wide bandgap of ZnO (3.2–3.4 eV) [19]. This fact limits the efficiency of solar light utilization because the UV region is less than 5% of the solar spectrum [20,21], while visible light is much safer to humans than UV light and makes up almost half of the radiation from the Sun. Thus, enhancing the efficiency of sunlight utilization is a crucial task, which can be addressed through the development of new photocatalysts with a visible-light response (e.g., metal sulfides [22,23], bismuth oxyhalides [24], and graphitic carbon nitride [25]) or modification of conventional photocatalysts to extend their action spectrum to the visible region [26].

TiO_2_ doping with nitrogen (TiO_2_-N) has emerged as a successful strategy of its modification that offers the extension of light absorption into the visible region (up 550 nm) and enhanced photocatalytic degradation activity toward various organic pollutants [27,28]. TiO_2_-N can be employed for modification of fabrics to provide them with self-cleaning properties under visible light. For instance, self-cleaning fabrics modified with TiO_2_-N were prepared via their impregnation with an alcohol mixture of TiO_2_-N nanosol, which was obtained by sol-gel method using titanium(IV) n-butoxide (Ti(OBu)_4_) [29], or titanium(IV) isopropoxide (Ti(^i^OPr)_4_) [30] and triethylamine ((C_2_H_5_)_3_N). These materials were thermally treated under air conditions at a temperature of no more than 150 °C because of the low thermal stability of the textile base [31]. However, highly active TiO_2_ photocatalysts are commonly formed at higher temperatures: above 300 °C for the treatment under air conditions [32] or 150–200 °C for hydrothermal treatment [33,34]. This means that the synthesis of highly active coating on the fabric surfaces cannot be realized due to the impossibility of treatment at high temperatures, and the development of low-temperature deposition methods is important.

Immobilization of well-crystalline photocatalyst particles onto the fabric surface is one of successful strategies for the preparation of photocatalytic materials with a high activity [35]. For instance, the fabric was impregnated with aqueous or alcoholic dispersions of TiO_2_-N, which was synthesized via high-temperature calcination of the prepared nanosol [36,37]. This fabrication method demonstrated superior photocatalytic activity of as-prepared materials relative to the methods described above [38]. However, the obtained photoactive fabric exhibited poor interfacial adhesion of the photoactive component to the fabric’s surface due to the lack of strong bonding between TiO_2_-N particles and textile functional groups on the surface of the fiber [39,40]. As a result, photocatalytic performance of material substantially decreased after washing procedures due to flushing of many titania particles. This problem can be solved by using a combined approach when the initially formed photocatalyst particles are fixed on the fabric using a binder, which provides strong bonding and increases coating stability [41,42]. Self-cleaning fabrics have been prepared using organic binders (e.g., carboxylic acids [43,44]), as well as inorganic compounds (e.g., silica [45]). However, carboxylic acids could be degraded during the photocatalytic oxidation process, thus resulting in a decrease in adhesion between photocatalytic particles and the surface of material and a decline of its photoactivity due to removal of particles from the fabric [46]. Inorganic substances can significantly improve the photocatalytic properties of fabric while being the binding material [47]. Addition of SiO_2_ into the impregnated composition was shown to enhance the mechanical stability of the photoactive coating by incorporating TiO_2_ into the silicon matrix [40,45], which also positively affected the rate of photocatalytic decolorization of dye stains [48]. Besides SiO_2_, amorphous TiO_2_ formed during the hydrolysis of titanium(IV) isopropoxide was used as a binder contributing to highly active cotton fabric with an uniform and stable photoactive coating [49].

In this study, we successfully prepared a self-cleaning textile based on a blended cotton/polyester fabric functionalized with TiO_2_-N nanoparticles using titanium(IV) isopropoxide as a binder and additionally decorated with Cu species. The photocatalytic ability of the material was investigated in oxidation of acetone vapor, degradation of DNA fragments of various lengths, and inactivation of PA136 bacteriophage and *Candida albicans* fungi under visible light and UVA radiation. The as-prepared self-cleaning fabric exhibited stability of photoactive coating and high performance in the inactivation of viruses and fungi. Additional modification of the material with Cu species substantially improved its self-cleaning properties, even in the absence of light.

## 2. Results and Discussion

Visible-light-active TiO_2_-N, known for its ability for complete mineralization of various pollutants [50], was employed as the photocatalytic component to obtain photoactive fabric with self-cleaning properties. TiO_2_-N was prepared via the pH-controlled precipitation of titania from an aqueous solution of TiOSO_4_ using ammonia as a precipitating agent, as well as a source of nitrogen, followed by the calcination of precipitate in air at an increased temperature. This method resulted in the formation of nanocrystalline anatase, which can absorb radiation in the region up to 540 nm due to the incorporation of nitrogen impurities from ammonia into the TiO_2_ lattice. The photoactive coating was deposited onto the fabric surface using the impregnation method. Titanium(IV) isopropoxide, which was hydrolyzed during the synthesis with a formation of TiO_2_ matrix, was used as a binder to anchor the TiO_2_-N particles onto the fabric surface. Since copper is known to enhance the photocatalytic properties of titania in both the pollutant degradation and the inactivation of biological objects [51], the textile material was additionally modified with Cu species by using copper acetate in the impregnating compound.

### 2.1. Material Characteristics

The SEM image shows that fibers of the initial fabric (**IF**) have a smooth surface (Figure 1a). The prepared **PF-TN** material exhibits a coating with fixed agglomerates of TiO_2_-N particles, which have a size of 1–5 µm (Figure 1b). The surface morphology of this coating is shown in Figure 1c: the observed lamellar structure can be attributed to the binder layer. The small particles in the layer can be attributed TiO_2_-N agglomerates with size of several hundred nanometers. Thus, the use of a titanium binder made it possible to reliably fix photocatalytic particles on the fabric fiber. Additional copper deposition did not change surface morphology (Figure 1d). This is to be expected, since when copper is applied in this way, it is distributed evenly and randomly, without leading to the formation of large particles, even on the surface of the titania photocatalyst [52]. Bright-field scanning transmission electron microscopy (BFSTEM) (Figure 1e) and energy dispersive X-ray (EDX) mapping (Figure 1f) images confirm this statement and show that copper is evenly distributed on the surface of coated fibers in the **PF-TN-Cu** material.

The XPS data confirmed the successful fixation of the photocatalytic particles on the surface of the initial fabric sample (**IF**), which did not have any peaks in the Ti2p photoelectron spectral region (Figure 2a). The presence of TiO_2_ on the **PF-TN** surface is clearly evident by the presence of two characteristic peaks at 458.8 and 464.4 eV, which can be attributed to Ti2p_3/2_ and Ti2p_1/2_ of Ti^4+^ state [53].

The surface Ti contents in **PF-TN** and **PF-TN-Cu** samples were determined as ca. 17% and 14%, respectively, considering the atomic scattering factors. The total Ti content determined by the ICP-OES method was 3.4 wt% and 2.9 wt% for **PF-TN** and **PF-TN-Cu**, respectively. A slight decrease in titanium amount in the **PF-TN-Cu** sample was observed on the surface as a result of the copper application procedure and a partial wash-out of TiO_2_ particles.

It is worth noting that no XPS signals are observed in the photoelectron N1s and Cu2p spectral regions of **PF-TN-Cu** material, although nitrogen and copper states are evident for the powdered Cu-modified TiO_2_-N photocatalyst (**TN-Cu**) by the means of the XPS method (see Appendix A in the Appendix A). This fact results from an extremely low content of these elements due to the predominance of fabric and TiO_2_ components in the composition of photoactive material. Nitrogen and copper contents in **PF-TN-Cu** are estimated as 0.01 and 0.03 wt%, respectively, and these values are much lower than the detection limit of the XPS method.

The XRD pattern of the **IF** sample had peaks of the initial fabric components only (Figure 2b). Thus, the high-intensity peaks at 15.2°, 16.6°, 20.5°, 22.7°, and 34.4°, which correspond to the $\left(1\bar{1}0\right)$, (110), (102), (200), and (004) planes of cellulose Iβ, were detected because cotton was the main component of the fabric used [54]. At the same time, peaks at 17.6° and 25.6° attributed to the (010) and (100) planes of polyester (JCPDS No. 50-2275) proved the initial fabric composition [55]. No evidence of the presence of spandex was observed possibly due to its small content in the material (less than 2 wt%) and the low intensity of the corresponding signals being overloaded by other fabric peaks [56]. The peaks described above were observed for all synthesized samples.

The surface of modification of initial fabric with TiO_2_-N resulted in the appearance of additional peaks in the XRD pattern of **PF-TN** at 25.5°, 38.2°, 48.2°, 54.2°, 55.2°, and 63.0° (Figure 2b). They can be attributed to the (101), (004), (200), (105), (211), and (204) planes of TiO_2_ anatase (JCPDS No. 21-1272) [57]. The XRD pattern of **PF-TN-Cu** material is completely consistent with the diffractogram of **PF-TN**, where the peaks described corresponded with the initial fabric and titanium dioxide. No copper compounds were observed due to their low content on the material’s surface, and they did not form crystallites. In the case of TiO_2_-N powder, the copper can be found in the forms of Cu^0^, Cu_x_O, Cu^2+^ [58], but on the surface of fabric it predominantly presents in the form of Cu_x_O [59].

UV–Vis diffuse reflectance spectra (Figure 2c) showed that all samples had two main regions of light absorption. The first significant decrease in the reflection below 400 nm can be attributed to polyester, which has an absorption edge at ca. 390 nm [60]. Another light absorption in the 500–750 nm range is caused by the presence of blue dye in the used fabric. An absorption in the range of wavelengths shorter than 450 nm increased for the **PF-TN** sample due to light absorption by the deposited TiO_2_-N [61]. The presence of Cu increased the absorption on the 350–500 nm for the **PF-TN-Cu** sample. Therefore, the results of analysis proved the successful formation of coating on the surface of cotton/polyester fibers.

### 2.2. Photocatalytic Oxidation

The self-cleaning properties were firstly investigated in a test reaction of acetone vapor oxidation under visible light and UVA radiation. The rate of CO_2_ formation was chosen as the measure of photocatalytic activity of samples (Figure 3). The photocatalytic activity of the **IF** sample did not exceed the detection limit and was mostly caused by the CO_2_ desorption from the fabric surface under irradiation. This confirms that non-modified fabric cannot oxidize such chemicals. The photocatalytic activity of **PF-TN** which equaled 0.1 μmol min^−1^ under visible light was caused by the presence of a photoactive component, namely TiO_2_-N, in its composition. The obtained value was significantly lower than that of **TiO_2_-N** powder (0.92 μmol min^−1^) used for fabric preparation. The activity reduction may be assigned to a low catalyst content in the fabric material (less than 4 wt% of titanium amount). The same behavior was observed for samples irradiated with UVA: the activity value was 0.36 μmol min^−1^ for the **PF-TN** sample and 1.39 μmol min^−1^ for the **TN** sample (TiO_2_-N powder).

Regardless of the fact that the visible-light activity of Cu-modified TiO_2_-N (**TN-Cu**) photocatalyst with Cu loading of 1 wt% is ~30% higher than the activity of the initial TiO_2_-N (**TN**) sample (Appendix A in the Appendix A), the activity of **PF-TN-Cu** material similarly equals to 0.1 μmol min^−1^ under visible light as for **PF-TN** (Figure 3). This means no noticeable effect of Cu species on the material activity under visible light. The activity of TiO_2_-N powder is known to increase after copper modification due to improving the transfer of photogenerated holes to the photocatalyst surface [58]. Probably, the concentration of Cu species on the surface of TiO_2_-N crystallites in the case of **PF-TN-Cu** is lower than in the case of powdered **TN-Cu** photocatalyst because many Cu species are absorbed on the surface of fabric fibers and amorphous titania binder. As a result, the surface concentration of Cu species on the surface TiO_2_-N crystallites is not high enough to substantially improve the separation of charges photogenerated in TiO_2_-N crystallites and enhance the overall photocatalytic activity of the material. An additional argument in favor of copper adsorption on fabric fibers is the evidence of improved adsorption of copper ions in the presence of hydroxyl groups on fibers [62,63]. Under UVA radiation, **PF-TN-Cu** exhibits a slight decrease in the activity compared to **PF-TN**, but the observed activity is still high enough for productive oxidation of chemicals.

Note that application of the titanium(IV) isopropoxide binder in the impregnating dispersion provides incorporation of the photocatalyst’s nanoparticles into the titania matrix and their attachment to the surface of the fabric fibers. As a result, the as-prepared materials have a good stability toward washing procedures (Appendix A). Another aspect shown in our previous paper [49] is that the TiO_2_-coated fabrics exhibit steady-state photocatalytic activity under long-term irradiation for days. All these factors confirm their potential for practical application as self-cleaning textile fabrics.

### 2.3. DNA Degradation

The contaminant degradation rate can significantly change when changing the pollutant studied from chemicals to biological objects. The rate of methylene blue and *Staphylococcus aureus* bacteria removal was shown to change under light on the surface of fabric modified with Mn-doped TiO_2_ nanoparticles [64]. Simple and representative biological contaminants are nucleic acids, but they are rarely used to characterize self-cleaning properties of materials. However, the presence of deoxyribonucleic acid (DNA) and ribonucleic acid (RNA) in the ambient air or on the surface of the material may interfere with obtaining representative results of any studies using highly sensitive microbiological methods such as PCR [65]. Thus, the study of nucleic acid (NA) degradation is of high importance.

The kinetics of nucleic acid degradation was studied using DNA fragments of various lengths under UVA radiation on the surface of the photoactive fabric (**PF-TN**) and initial fabric (**IF**) as the reference. The change in DNA concentration was calculated according to Equation (1) using the Cq values determined by the PCR method. The obtained kinetic curves of each fragment are shown in Figure 4a–d. DNA concentration remained constant on the **IF** surface and matched the initial DNA fragments’ concentration since the used unmodified fabric did not have any anti-contamination properties. An effect of UVA light on the DNA structure was observed on the **IF** surface (IF under UVA, Figure 4). The DNA concentration decreased by less than tenfold even after 360 min of UVA irradiation. This could be caused by DNA damage under ultraviolet irradiation even in the absence of titanium dioxide [66]. We observed the same pattern in our previous study [67]. Therefore, the stability of DNA on the initial fabric surface and a slight decrease in its concentration under irradiation highlights the ability of NA molecules to be stable on the fabric surface. This makes it possible to transfer them from contaminated fabric to other materials and surfaces.

A much more noticeable influence of UVA light was observed for samples with photoactive coating. A small decrease in DNA concentration occurred on the non-irradiated material **PF-TN** (Figure 4): the decrease in concentration was about 10 % at point 0 min, and then concentration monotonically declined during 360 min and reached the value observed for the **IF** under UVA light. That may be caused by the growth of adsorption characteristics of fabric after modification with TiO_2_-N.

In the case of the UVA-irradiated **PF-TN** sample, the DNA concentration markedly fell during the initial 120 min, and then it slowly decreased the next 240 min to the 1000-fold decrease relative to the initial value (Figure 4). The significant fall of the DNA amount is caused by the production of reactive oxygen species (ROS), especially hydroxyl radicals (OH^●^), on the UVA-irradiated surface of titania, which can oxidize the nucleic acids [68]. Such oxidation of DNA leads to a decrease in its concentration due to the oxidative damage of nitrogenous bases as well as the accumulation of both single-strand breaks (SSBs) and double-strand breaks (DSBs) in DNA molecules followed by its complete degradation to carbon dioxide, phosphates, and nitrogen compounds according to the mechanisms described in our previous paper [67]. It is worth noting that an increase in the fragment length led to growth in the removal rate. The DNA concentration decreased by 1600, 3200, 7000, and 8100 times for the DNA lengths of 71, 126, 226, and 454 bp, respectively (Figure 4a–d). This is a consequence of the non-selective nature of TiO_2_-mediated photocatalytic oxidation under UVA irradiation [69] and, therefore, the increased oxidation probability for molecules with increased length [67].

Also, DNA degradation was studied using a modified method with three-point activity measurement under visible light: this will still allow obtaining relevant data on the DNA concentration changes, while requiring fewer resources. For this, the volume of applied DNA mixture was increased from 25 to 100 µL and the control experiment under UVA irradiation was performed to check out the effect of the contaminant volume. The DNA concentration change on the material surface under visible light was investigated after 3 and 8 h of irradiation to check out that the degradation pattern for DNA was the same for both the irradiation types as it was for chemical contaminations [70]. The relative decrease in DNA concentration was calculated by normalizing the DNA concentration on irradiated materials to those that were kept in the dark. The obtained threshold cycle values (Cq) are shown in Appendix A. It was shown that the DNA concentration on the surface of **PF-TN** decreased 210–360 times compared to the initial level after 3 h of UVA irradiation (Figure 5a). For the degradation kinetic curves under similar conditions for DNA fragments with lengths of 71–226 bp (Figure 4), these values were ca. 320–1510. This shows that the DNA degradation degree depends on the initial DNA concentration and that the fourfold increase in the amount of applied DNA (100 µL instead of 25 µL) decreases DNA degradation degree within the same period. Increasing the irradiation time up to 8 h allowed us to increase the DNA degradation degree significantly (DNA concentration 900–3000 times less than initial). So, the modified approach is shown to give a valid result by increasing the irradiation time.

The DNA concentration on the surface of **IF** material decreased 2–16 times after 8 h of irradiation under visible light which could be mostly caused by the thermal effect of the radiation from the visible-light LED on the DNA structure (Figure 5b). The photoactive fabric **PF-TN** significantly decreased the DNA amount on its surface: reduction by 170, 440, and 900 times after 8 h of lighting for DNA fragments with lengths of 71, 126, and 226 bp was observed, respectively. This confirms the dependence of the DNA degradation rate on nucleic acid length within the same mechanism for this process. Herewith, the obtained DNA degradation rate under 450-nm irradiation was 3–5 times less than that under UVA irradiation, which is consistent with the photocatalytic activity observed under visible and UVA LEDs (Figure 3). The results obtained show the ability of photoactive fabric **PF-TN** to provide DNA degradation on its surface under both UVA and visible light.

### 2.4. Antiviral Activity

The ability of self-cleaning photoactive materials to degrade the membranes and cell walls of microorganisms, including endospores, fungi, bacteria, and viruses, causing their death, holds a special interest on such materials because of the attractiveness for practical use [39,71]. Although numerous studies of photoactive materials and their antibacterial activity are presented in the literature [72,73], very few papers are dedicated to their antiviral activity [74]. Those publications are mostly focused on the destruction of enveloped viruses, such as influenza virus [42], SARS-CoV-2 coronavirus [75], murine coronavirus (MHV-3) [74], because these viruses can replicate themselves quickly in human society. It was proved that photoactive materials effectively inactivate influenza virus and completely destroy its structure down to RNA under UVA irradiation [42].

Non-enveloped viruses lack the additional lipid bilayer, and are therefore essentially different in their structure. The inactivation of this type of virus, e.g., human adenovirus (HAdV-5) [74] and human norovirus [76], has been also studied. Nevertheless, bacteriophage inactivation is understudied as yet. For this reason, the non-enveloped bacteriophage PA136 from the *Myoviridae* family containing a double-stranded DNA was selected as an object for this research [77].

The virus inactivation ability of prepared samples was studied as control, because it is crucial to prevent possible cell infection with any viruses. For this purpose, the virus suspension was applied to the surface of **IF** and **PF-TN** materials and exposed to radiation with visible light. A slight decrease in virus concentration on both the initial **IF** and photoactive fabric **PF-TN** was observed even in the absence of irradiation (Figure 6a): from 1.6 × 10^7^ to 4.4 × 10^6^ PFU mL^−1^, which is a four-fold decrease. This is about the same decrease as for the virus sample placed on the glass surface under the same conditions (≈3.6 times, Appendix A). This can be associated with the damage of the virus itself in the environmental conditions.

A slight reduction in bacteriophage concentration was noticed on the surface of initial fabric **IF** under visible light irradiation (Figure 6a): concentration decreased ca. 4.3 times under visible light (namely from 1.6 × 10^7^ to 3.7 × 10^6^ PFU mL^−1^), which is presumably caused by heating of the virus-supporting fabric under visible irradiation. In the case of the irradiated photoactive **PF-TN** sample, the phage concentration slightly decreased during the initial 2 h. Then, after 3 h of the experiment, a sharp decrease in the virus concentration occurred with inactivation of ca. 99.9% of virions. The observed induction period can be related to the need for accumulation of enough damage to inactivate the viral particle. Complete inactivation with a decrease in the virus amounts down to the detection limit occurred after 6 h. The same trend was observed in materials irradiated by UVA (Appendix A). However, the phage concentration decreased significantly faster (more than 99.8% in 2 h), which may be associated with a higher photoactivity of materials under UVA described earlier (Figure 3). Thus, the synthesized materials demonstrate antiviral activity both under visible and UV irradiation, reducing the virus amount by more than 99.8% during 2–3 h of irradiation.

### 2.5. Effect of Cu Species on DNA Degradation and Antiviral Activity

The copper effect on antiviral properties of the materials obtained was additionally studied using the **PF-TN-Cu** sample (Figure 6b). The phage concentration decreased more than 10 times after 6 h even in the absence of radiation (inactivation degree more than 96% of virions instead of 53% earlier), which is caused by antimicrobial properties of copper itself [78]. In the case of irradiated **PF-TN-Cu** material, the decrease in the phage concentration after 2 h was shown to be 99.7 and 98.6% under visible and UVA radiation, respectively (Appendix A). It is worth noting that phage concentration was almost unchanged in the absence of copper (Figure 6a) within the initial 2 h. So, this means that copper application onto the photoactive coating decreased the observed induction period. At the same time, the Cu addition did not affect the time of complete inactivation: the kinetic curves for materials with and without copper addition matched completely after more than 3 h of irradiation (Appendix A). Therefore, we may conclude that copper decreased the induction time due to its antimicrobial properties, while the overall antiviral effect exists because of catalytic oxidation on the irradiated photoactive surface.

In addition to virus inactivation process, the Cu-addition effect was studied for DNA degradation too. In the absence of copper, the DNA concentration hardly changed on unirradiated samples and on **IF** even under irradiation. However, a two-fold decrease in DNA concentration was observed on the **PF-TN** sample after 6 h of irradiation with visible light, confirming the existence of photocatalytic antiviral properties (Figure 6c). One can see that phage inactivation occurred much faster than DNA destruction: this is obvious because the bacteriophage DNA is protected by capsid proteins from oxidative effects (including the effect of the photocatalyst).

The degradation of phage structure also occurred over the samples considered under UVA irradiation (Appendix A), which aligned well with the virus inactivation data and photocatalytic oxidation activity values. The **PF-TN-Cu** material led to an increase in the rate of DNA degradation by more than order of magnitude if compared to the **PF-TN** sample which was not modified with copper (Figure 6d and Appendix A). Copper is known to be an effective antimicrobial agent, which can damage viral genomic DNA due to synergistic action of copper ion attack (binding and cross-linking between the strands of NAs) and ROS generation reaction [79]. Therefore, the fundamental antiviral effect of the materials considered is caused by photocatalytic oxidation over TiO_2_-N with irradiation by UVA or visible light while the copper presence is required for effective removal of contaminants on the material surface in the initial stages of the process.

### 2.6. Antifungal Properties

Fungi are other harmful biological contaminants, since they can cause a wide range of diseases [80,81]. In contrast to viruses, the fungi have a thick and durable cell wall protecting them from the negative environmental impact [82]; therefore, a much longer period is required for their removing from the surface of any material compared to the Gram-negative bacteria [83]. So, the antifungal properties of synthesized self-cleaning materials are strongly important characteristics, which are usually studied using *Aspergius niger* [84,85], *Aspergillus flavus* [86], *Penicillium chrysogenum* [86], and the most common object for studying the materials—the diploid fungus *Candida albicans* [87,88,89]. Therefore, this was the one used as the sample to study and compare the data obtained with the results of previous studies. Antifungal experiments were carried out on the initial fabric **IF** and the copper-modified photoactive material **PF-TN-Cu**, which had the highest activity and contaminant removal effect.

It was shown that the **IF** did not adsorb fungi cells on its surface at all and the cell concentration completely coincided with initial one (Figure 7, point 0 min). Furthermore, the concentration insignificantly changed from 1.1 × 10^5^ to 0.8 × 10^5^ and to 0.6 × 10^5^ CFU mL^−1^ during 150 min in the dark and under UVA irradiation, respectively (Figure 7a, grey and black lines). This means that *Candida albicans* remained viable on the **IF** surface in the dark as well as under UVA light. Perelshtein et al. showed equality of *Candida albicans* concentration after incubation on a cotton surface for 1 and 3 h without light [90] meaning the cells were viable on the surface of non-modified fabrics. Additionally, the cell concentration was not changed on the surface of UVA-irradiated pristine fabric because UVA irradiation is ineffective for destruction of this type of contaminants [91].

The presence of photoactive coating in the **PF-TN-Cu** sample significantly increased the *Candida albicans* adsorption (Figure 7a). Such an effect was noted also during the bacteria destruction studies that proved the adsorption of more than 70% of *E. coli* cells occurred on the cotton surface (Appendix A). When the fungus cells were kept on the photoactive fabric surface during 60–90 min in the dark, the concentration of cells evenly decreased down to 9.6 × 10^2^ CFU mL^−1^. This behavior of the kinetic curve can be explained by the presence of copper, which has antimicrobial properties [78]. However, cell count remained about 3.5 × 10^2^–5.2 × 10^2^ CFU mL^−1^ even after 120 min and was stable during the next 2 h (Figure 7b). So, neither the high adsorption by TiO_2_ nor the antimicrobial properties of Cu were enough to inactivate the fungi cells completely. However, the light irradiation improved the antifungal properties of **PF-TN-Cu** material. Thus, the cell concentration monotonically decreased down to the detection limit (less than 10 CFU mL^−1^) under UVA irradiation during 150 min, which was caused by the formation of ROS on the surface of TiO_2_-coated material [92]. Such a high antifungal ability is already known [93,94]. As in the case of UVA irradiation, the *Candida albicans* was also inactivated on the **PF-TN-Cu** photoactive material irradiated with visible light (Figure 7b). The inactivation period under visible light is only 120 min compared to 150 min under UVA, which might be caused by the 20-times higher irradiation density from the visible-light LED. Indeed, a significant decrease in CFU of *Candida auris* was observed with the rise of the visible light dose [95]. As shown by Lozano-Rosas et al., the presence of Cu on the TiO_2_ surface reduced the number of *Candida albicans* cells and the visible light irradiation enhanced antifungal properties of photocatalyst [96]. Therefore, the obtained self-cleaning fabric materials can inactivate even such high-resisting objects because of the occurrence of the TiO_2_-N photoactive layer and copper with pronounced antimicrobial properties.

The obtained photoactive fabrics functionalized with TiO_2_-N can provide degradation of various biological contaminants on their surface under visible light or UVA radiation. The self-cleaning ability of materials under visible light shows promise for their usage in creating fabric products, such as medical coats and curtains, for medical and scientific institutions that require high levels of environmental cleanliness. The possibility of their practical application in the mentioned areas is also provided by the low cytotoxicity of used functional components toward human skin cells, even with prolonged contact [37,97]. Although photoactive fabrics functionalized with a nanocrystalline photocatalyst have a potential risk of leaching the photocatalyst’s nanoparticles [98], the proposed preparation technique solves this problem via use of the titanium(IV) isopropoxide binder, which forms a titania matrix and incorporates the photocatalyst’s particles. Thus, they do not present as single nanoobjects, and the as-functionalized photoactive fabrics exhibit low environmental impact.

## 3. Materials and Methods

### 3.1. Synthesis of Materials

The pieces of a blue-colored fabric consisting of 55% cotton, 43% polyester, and 2% spandex (155 g m^−2^, Jiangsu Xinghe Textile Co., Ltd., Wujiang, China) were used as substrates to obtain self-cleaning textile samples. All fabric pieces were preliminarily washed in an aqueous solution of Triton X-100 nonionic surfactant (Helicon, Moscow, Russia) and laundered with deionized water and isopropyl alcohol (CH_3_CH(OH)CH_3_, AO EKOS-1, Moscow, Russia) followed by drying at 60 °C. Titanium(IV) isopropoxide (Ti(OC_3_H_7_)_4_; >98%; Acros Organics, Waltham, MA, USA) dissolved in isopropyl alcohol was employed to anchor the photocatalyst’s nanoparticles on the fabric surface. Titanium(IV) oxysulfate dihydrate (TiOSO_4_∙2H_2_O, AO Vekton, Moscow, Russia) and aqueous solution of ammonia (NH_4_OH, 25%, AO Reachem, Moscow, Russia) were used to synthesize nitrogen-doped titanium dioxide (TiO_2_-N) which served as the visible-light photocatalyst. Copper(II) acetate (Cu(OAc)_2_) monohydrate (Cu(CH_3_COO)_2_∙H_2_O, 99%, AO Vekton, Russia) was employed to modify the material’s surface with copper species. All reagents were high purity grade and used without further purification.

### 3.2. TiO_2_-N Synthesis

Nitrogen-doped titanium dioxide (**TN**) was synthesized according to our previously published technique [58], in which the titanium(IV) oxysulfate dihydrate and an aqueous solution of ammonia were added dropwise into deionized water under a constantly maintained pH of 7 (Figure 8a). After aging for two days under continuous stirring, the sediment was washed with deionized water and then calcined under air atmosphere in an electric oven at 450 °C for 3 h. The as-obtained obtained powder was milled using a Retsch MM500 nano mixer mill (Retsch GmbH, Haan, Germany) using zirconium oxide balls (0.1 mm) and isopropyl alcohol, followed by drying at 70 °C.

### 3.3. Synthesis of Photoactive Fabric

Photoactive fabric modified with TiO_2_-N nanoparticles (**PF-TN**) was prepared by the impregnation method using titanium(IV) isopropoxide as a binder [42]. The scheme of **PF-TN** synthesis is shown in Figure 8b. Briefly, a piece of the initial fabric (**IF**) was impregnated with a suspension of **TN**, titanium(IV) isopropoxide, and isopropyl, pressed, and dried at room temperature. Actual quantities of used components are listed in Table 1. After that, the second layer of the photoactive coating was deposited using the same technique to increase the thickness of photoactive coating and, consequently, the photoactivity of the material. Finally, the material was treated with water steam, dried at 70 °C, and washed with water followed by drying at 70 °C.

The copper-modified material (**PF-TN-Cu**) was synthesized by addition of copper(II) acetate monohydrate to the impregnated composition described above. The amount of added Cu(OAc)_2_ was estimated on the basis of achieving a copper loading of 1 wt% relative to the mass of the TiO_2_-N component (Table 1) because it was regarded as an optimum value [58]. The other parameters of the synthesis procedure were kept the same. Table 1 shows the composition of each sample, and the quantities of the components used.

### 3.4. Material Characterization

A JSM-6460 LV microscope (JEOL, Tokyo, Japan) was used to investigate the surface morphology of samples by scanning electron microscopy (SEM). Scanning transmission electron microscopy (STEM) images were obtained in the bright-field (BF) mode using a Regulus 8230 field-emission scanning electron microscope (FE-SEM, Hitachi, Tokyo, Japan) at an accelerating voltage up to 30 kV. Element distribution was scanned using a Tescan Solaris microscope (TESCAN GROUP, Brno-Kohoutovice, Brno, Czech Republic). X-ray powder diffraction (XRD) analysis was performed using an ARL X’TRA diffractometer (Thermo, Basel, Switzerland) equipped with a CuKα radiation source and a Mythen2 X 1D linear detector (Detris, Baden-Daettwil, Switzerland) to determine the phase composition. The UV–Vis diffuse reflectance spectra (DRS) of the prepared materials were recorded in the range of 250–850 nm with a resolution of 1 nm using a Cary 300 UV–Vis spectrophotometer (Agilent, Santa Clara, CA, USA) equipped with a DRA-30I diffuse reflectance accessory; polytetrafluoroethylene was used as a reflectance standard. The surface state of Ti was determined by X-ray photoelectron spectroscopy (XPS) using a SPECS spectrometer (SPECS Surface Nano Analysis GmbH, Berlin, Germany) equipped with a PHOIBOS-150-MCD-9 analyzer (AlK_α_ radiation, hν = 1486.6 eV, 150 W). The peak positions of Au4f_7/2_ and Cu2p_3/2_ with binding energies equal to 84.0 eV and 932.67 eV, respectively, were used for instrument calibration. Ti content in the samples was measured by inductively coupled plasma optical emission spectroscopy (ICP-OES) using an OPTIMA 4300 DV spectrometer (PerkinElmer, Springfield, IL, USA).

### 3.5. Photocatalytic Activity

The ability of samples to oxidize volatile organic compounds under exposure to light was investigated according to the test method described in our previous papers [28,49]. Briefly, each fabric sample (9 cm^2^) was placed into the continuous-flow photoreactor and irradiated sequentially with two light-emitting diodes (LEDs) providing blue light (λ_max_ = 450 nm) or UVA light (λ_max_ = 365 nm). In each case, the sample was located directly under the LED at a distance of 9 cm, and the measured irradiance of the sample was 160 mW cm^−2^ and 10 mW cm^−2^ for blue light and UVA light, respectively. The powdered samples were distributed on a glass surface of 9 cm^2^ to obtain an area density of 30 mg cm^−2^. Acetone vapor (CH_3_COCH_3_, Mosreaktiv LLC, Moskow, Russia) with a concentration of 30–32 μmol L^−1^ was used as a testing oxidizing compound. Volume flow rate of humidified air (relative humidity of 20%) was 0.065 L min^−1^. The amount of carbon dioxide (CO_2_) formed as the final product of complete oxidation of acetone was determined using in situ IR spectroscopy on a Nicolet 380 FTIR spectrometer (Thermo Scientific, Waltham, MA, USA) in the range of 2200–2400 cm^−1^ related to the stretching vibrations of C–O [99]. The photocatalytic activity of each sample was calculated by multiplying the concertation of evolved CO_2_ and the flow rate.

### 3.6. Degradation of Biological Contaminants

Degradation of biological contaminates on the surface of fabric materials was studied using the fragments of DNA molecules of various lengths, PA136 virus bacteriophage, and *Candida albicans* fungi under exposure to radiation from a low-pressure mercury lamp (AeroLife LLC, Moscow, Russia) providing UVA light (λ_max_ = 365 nm, 3 mW cm^−2^) or two LEDs providing UVA light (λ_max_ = 370 nm, 5 mW cm^−2^) and blue light (λ_max_ = 450 nm, 30–80 mW cm^−2^), respectively. The samples were located directly under the light source at a distance of 11–15 cm, and the actual values of measured irradiance are mentioned in the corresponding sections below. All experiments were carried out at a room temperature of 23–25 °C and relative humidity of 20% simultaneously using photoactive fabric materials and the initial fabric as a control. In addition, a part of the samples was irradiated with light, while another part was kept in the dark to estimate the self-cleaning ability of materials without light and to investigate the effect of irradiation on the self-cleaning properties. The zero-time point was used to illustrate the adsorption properties of materials. Each experimental point in the kinetic plots was the result of averaging at least three independent repeats, and the experimental error was evaluated as the standard deviation.

#### 3.6.1. DNA Degradation

The investigation of nucleic acid degradation was carried out with a mixture of four DNA fragments, namely HSPA8 (454 base pairs (bp)), GAPDH (226 bp), 28-2.2 (126 bp), and 18-1702 (71 bp). Initial concentration ($N^{0}$) of each fragment was adjusted to 0.125 μg mL^−1^. The kinetic curves of DNA concentration were determined by the polymerase chain reaction (PCR) method. An amount of 25 µL of DNA mixture was placed on the surface of a 1 cm^2^ piece of fabric and irradiated with UVA (5 mW cm^−2^) for 360 min. Then, the remaining DNA was extracted by the fabric incubation in a 1 mL of 0.01 M Tris-HCl (pH = 8.0, Biolabmix LLC, Novosibirsk, Russia) for 30 min, then the obtained samples were kept at 4 °C. The specific primers (Table 2) were added into the washed solution to determine the amount of DNA. The 28-2.1 PCR fragment corresponding to the sequence from 4502 to 4548 nucleotide of 28S ribosomal RNA was employed as an internal control to evaluate the presence of inhibitors. The threshold cycle value (Cq) was measured using a LightCycler^®^ 96 System and LightCycler 96 software version 1.1.0.1320 (Roche Diagnostics, Rotkreuz, Switzerland). The PCR protocol is described in detail in the Appendix A. The experimental Cq value was used to quantify the decrease in DNA concentration for all samples:(1)$$\log(\frac{N}{N^{0}})={log(2}^{-(Cq-{Cq}^{0})}),$$
where N is the DNA concentration on the material surface, $N^{0}$ is the initial DNA concentration, Cq is the threshold cycle value of the sample, and ${Cq}^{0}$ is the threshold cycle value of initial DNA.

The rate of DNA degradation due to photocatalytic oxidation under UVA and visible light was studied using the method described above with some corrections. The initial volume of DNA solution containing GAPDH, 28-2.2, and 18-1702 fragments was 100 µL and the material was irradiated with UVA (5 mW cm^−2^) or blue light (80 mW cm^−2^) for 0 h, 3 h, and 8 h. The experimental Cq of the sample irradiated with light and the one kept without light were marked as Cq_with light_ and Cq_without light_, respectively. The decrease in DNA concentration (*E*) showing the rate of DNA degradation was evaluated as follows using the obtained ΔCq values (see Appendix A):(2)$$∆C_{q}=C_{q\;\mathrm{with}\mathrm{light}}-C_{q\mathrm{without}\mathrm{light}},$$(3)$$E=2^{{∆C}_{q}}$$

#### 3.6.2. Antiviral Activity

The antiviral activity of materials was investigated using PA136 bacteriophage (phage) from the collection of extremophilic microorganisms and typical cultures (EMTCs) of the Institute of Chemical Biology and Fundamental Medicine of the Siberian Branch of the Russian Academy of Sciences (ICBFM SB RAS). This bacteriophage was preliminarily isolated in the Shemyakin-Ovchinnikov Institute of Bioorganic Chemistry of the Russian Academy of Sciences. It is a non-enveloped virus containing genomic double-stranded DNA and multiplying in *Pseudomonas aeruginosa* cells. Pieces of fabric samples 4 cm^2^ in size were placed in a Petri dish followed by adding 1 mL of bacteriophage suspension with concentration of 10^7^ plaque-forming units per mL (PFU mL^−1^) in 0.9 wt% saline solution (Renewal, Novosibirsk, Russia) onto the sample surface. Additionally, a bacteriophage suspension of equal value was placed in a Petri dish without samples to control the phage viability under experimental conditions (Appendix A). Then, the samples were placed on a cooling agent to eliminate overheating, covered with quartz glass to avoid evaporation, and irradiated with UVA (5 mW cm^−2^) or blue light (60 mW cm^−2^) for 0–6 h. Probes of 60 µL were taken every hour after mixing the suspension by pipetting to analyze bacteriophage infectivity and changes in DNA concentration.

Bacteriophage inactivation was studied using Pseudomonas aeruginosa bacteria from the collection of EMTCs of the ICBFM SB RAS by double-layer plaque assay (Gratia method) [100]. The concentration of the surviving phage was determined by titration with a tenfold step up to the fifth dilution in saline solution. An amount of 5 µL of each dilution of the solution was placed on agar with bacteria culture and incubated at 37 °C for 18–20 h. After that, phage concentration was calculated using the resulting plaque amount. The phage concentration (K) in PFU mL−1 was calculated as follows [101]:(4)$$K=\frac{a\cdot 1000\cdot 10^{n}}{V},$$
where a is the number of negative colonies in Petri dish in PFU, V is the volume of the seeded sample in mL, *n* is the dilution number of the preparation, for which the phage particles are calculated.

DNA concentration change on the material surface was investigated simultaneously with the virus inactivation study to prove the destruction of all bacteriophage structures. To accomplish that, special primers (Table 2) were added to the wash-off taken before PCR. Then, Cq was determined by the PCR method. The DNA amount was determined based on the linear Cq and log of DNA phage concentration dependency (Appendix A). It was plotted based on the correlation of the initial concentration of bacteriophage suspension in the range of 5 × 10^2^–5 × 10^9^ PFU mL^−1^ (1 PFU mL^−1^ equals 1 DNA copy in mL^−1^) and its measured Cq value. This allowed us to determine the DNA concentration in the samples from the Cq under similar conditions.

#### 3.6.3. Antifungal Properties

*Candida albicans* ATCC 10231 was used to study the antifungal properties of the materials. An amount of 10 mL of cell suspension with concentration of ca. 5 × 10^5^ colony-forming units per mL (CFU mL^−1^) of the *Candida albicans* night culture were deposited on samples of 1 cm^2^, placed in a Petri dish, then covered with a quartz glass to prevent the evaporation. The Petri dishes were placed on a cooling agent to eliminate the heating effect of the optical irradiation. The samples were irradiated by UV light from a mercury lamp (320–400 nm, 3 mW cm^−2^, AeroLife, Moscow, Russia) or blue light (30 mW cm^−2^) for 0–240 min. Then, every sample fragment was placed in 5 mL of 0.9 wt% saline solution (Renewal, Russia) for 2 h to allow cell desorption, after which it was stirred for 15 s. The number of cells was evaluated using the 10-fold dilution method [102]. An amount of 100 μL of aliquot was picked and diluted in 0.9 wt% saline solution, then spotted on Sabouraud agar (bioMérieux, Craponne, France) and incubated at 37 °C. After 48 h, colonies were counted to define the CFU mL^−1^.

## 4. Conclusions

Photoactive textile material based on blended cotton/polyester fabric was successfully synthesized via a simple impregnation method using TiO_2_-N nanoparticles as a visible-light-active photocatalyst and titanium(IV) isopropoxide as a binder. The addition of this binder, which was hydrolyzed during the synthesis with a formation of amorphous TiO_2_ matrix, allowed for anchoring TiO_2_-N particles onto the fabric surface. The as-prepared self-cleaning fabric exhibited stability of photoactive coating and high photocatalytic performance in the oxidation of volatile organic compounds and the degradation of biological contaminants under both UVA and visible light due to the presence of TiO_2_-N in the composition. The possibility of nucleic acid degradation under light was studied using DNA fragments of various lengths. The kinetic curves of DNA concentration changes showed that the concentration of each DNA fragment rapidly decreased by 10^3^ times during 2–3 h on the surface of the photoactive material under UVA irradiation. This proved the high efficiency of NA degradation under light. The presence of TiO_2_-N photocatalyst allowed us to decompose DNA molecules under visible light too. The rate of DNA degradation under UVA was higher than under visible light, which agreed with the photocatalytic activity of these materials in the oxidation of acetone vapor. Study of the degradation of PA136 virus bacteriophage confirmed the antiviral properties of the synthesized material. The virus concentration decreased by more than 99.8% under irradiation with visible light or UVA for 2–3 h. The addition of copper(II) acetate into the impregnating compound shortened the induction period and enhanced the antiviral activity of the material, even in the absence of light, thus showing that Cu species can substantially improve the self-cleaning properties of photoactive textiles. The kinetic curves of the *Candida albicans* cell concentration on the photoactive surface demonstrated a rapid decrease in fungi amount under UVA and visible light. It was proved that these self-cleaning fabric materials can inactivate even such resistant biological objects because of the both photoactive layer with TiO_2_-N and Cu species, which themselves have antimicrobial properties.

## Figures and Tables

**Figure 1 ijms-26-07550-f001:**
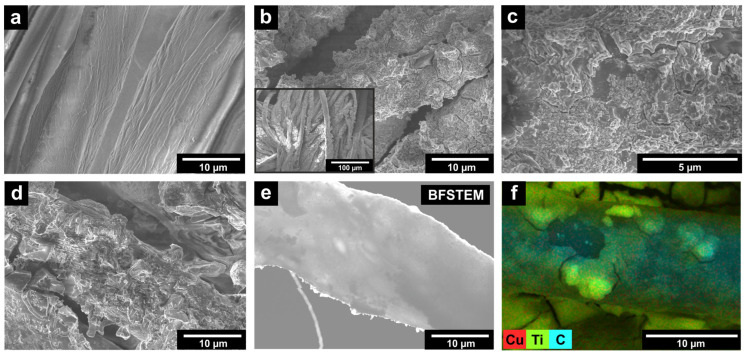
SEM micrographs of (**a**) initial fabric (**IF**) and prepared photoactive (**b**,**c**) **PF-TN** and (**d**) **PF-TN-Cu** materials; (**e**) bright-field STEM image of coated fiber in **PF-TN-Cu** material; (**f**) copper distribution in **PF-TN-Cu** material according to EDX analysis.

**Figure 2 ijms-26-07550-f002:**
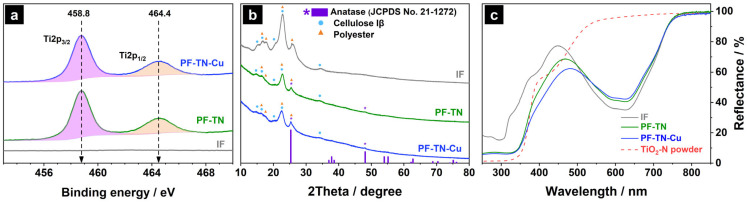
(**a**) XPS spectra in the Ti2p region, (**b**) XRD patterns, and (**c**) UV–Vis diffuse reflectance spectra of the initial fabric (**IF**), photoactive fabric modified with TiO_2_-N (**PF-TN**), and photoactive fabric modified with TiO_2_-N and Cu species (**PF-TN-Cu**).

**Figure 3 ijms-26-07550-f003:**
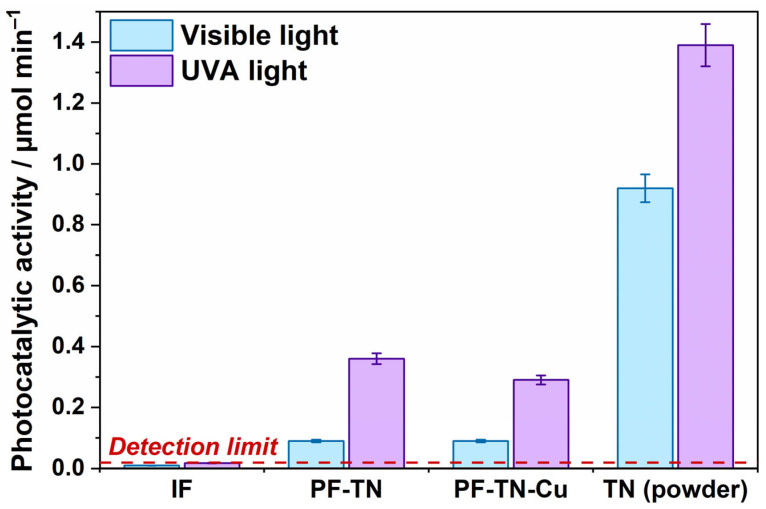
Photocatalytic activity of synthesized fabric materials and **TN** powder under visible light and UVA radiation.

**Figure 4 ijms-26-07550-f004:**
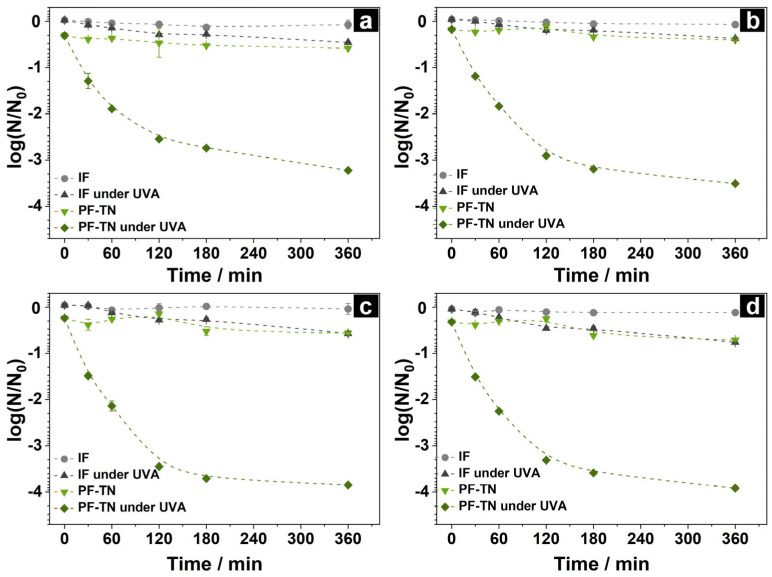
Kinetic curves of DNA concentration change under UVA irradiation on the surface of initial fabric (**IF**) and photoactive fabric with TiO_2_-N (**PF-TN**) using DNA fragments with lengths of (**a**) 71 bp, (**b**) 126 bp, (**c**) 226 bp, and (**d**) 454 bp.

**Figure 5 ijms-26-07550-f005:**
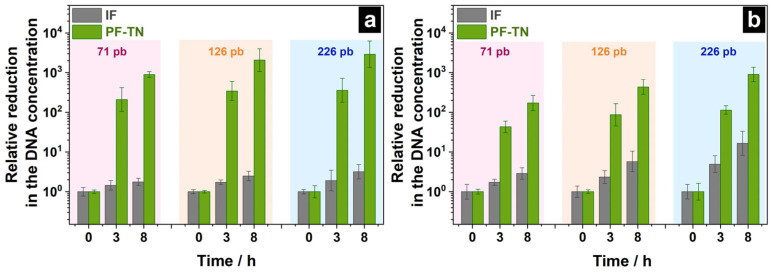
Relative reduction in DNA concentration on the surface of initial fabric (**IF**) and **PF-TN** material under (**a**) UVA and (**b**) visible light.

**Figure 6 ijms-26-07550-f006:**
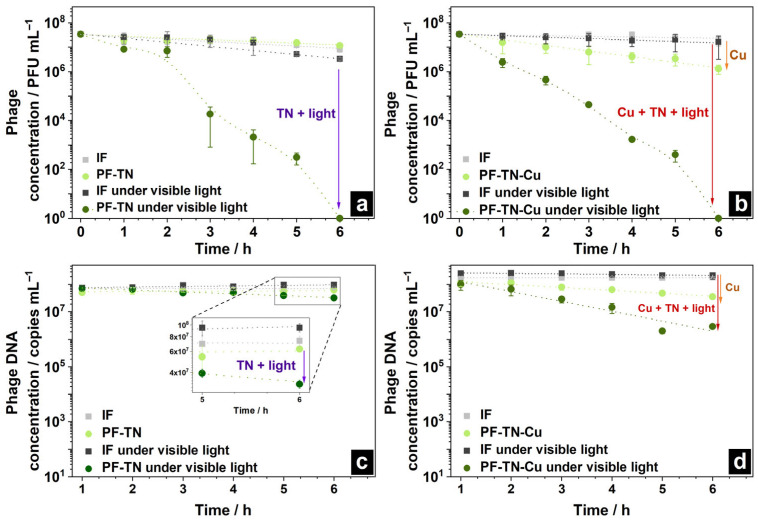
Change in phage concentration on the surface of (**a**) **PF-TN** and (**b**) **PF-TN-Cu** materials; change in its DNA on (**c**) **PF-TN** and (**d**) **PF-TN-Cu** surfaces under visible light.

**Figure 7 ijms-26-07550-f007:**
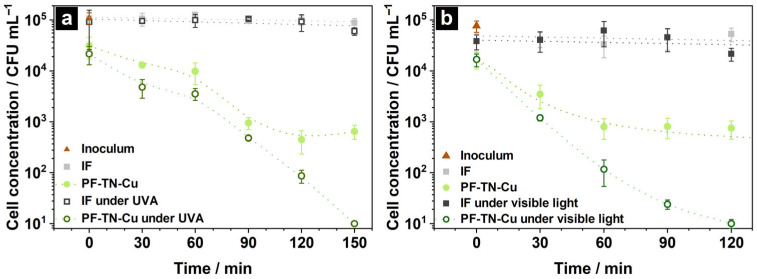
Kinetic changes in cell concentration of *Candida Albicans* on the surface of initial fabric **IF** and photoactive material **PF-TN-Cu** under (**a**) UVA irradiation and (**b**) visible light.

**Figure 8 ijms-26-07550-f008:**
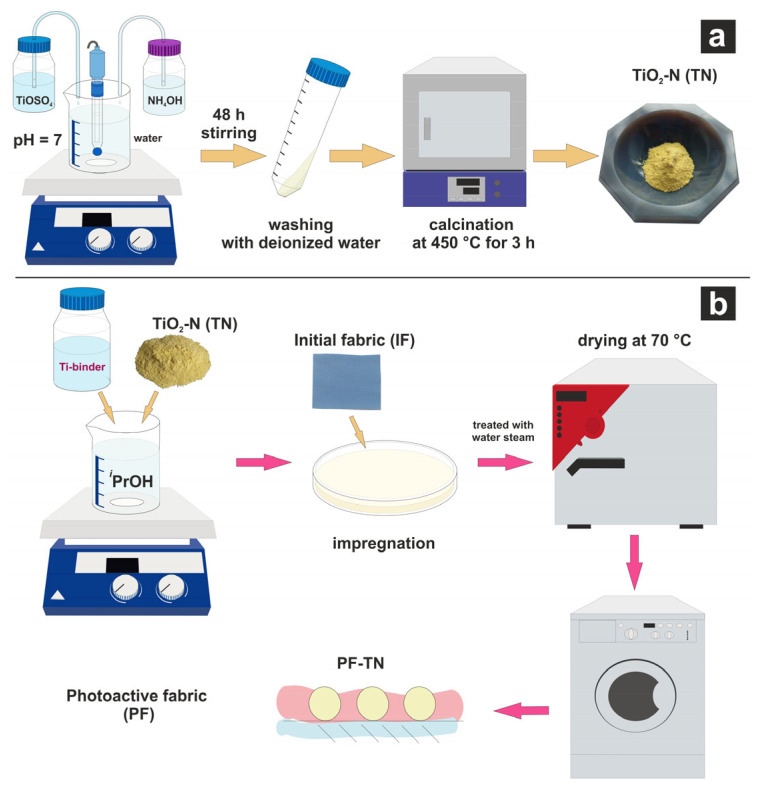
Synthesis schemes of (**a**) TiO_2_-N photocatalyst (**TN**) and (**b**) photoactive fabric (**PF-TN**).

**Table 1 ijms-26-07550-t001:** Composition of the suspension used for impregnating fabric pieces.

Sample	Composition of Impregnation Suspension			
	TiO_2_-N, g L^−1^	Ti Binder, vol%	^i^PrOH, vol%	Cu(OAc)_2_, g L^−1^
IF	–	–	–	–
PF-TN	10	5	95	–
PF-TN-Cu	10	5	95	0.28

**Table 2 ijms-26-07550-t002:** Sequences of the forward (F) and reverse (R) primers for each DNA fragment and DNA of PA136 bacteriophage.

Fragment	Primer Sequence	
HSPA8	F	5′-ACTGAACGGTTGATCGGTGA-3′
	R	5′-AGATGAGCACGTTTCTTTCT-3′
GAPDH	F	5′-GAAGGTGAAGGTCGGAGT-3′
	R	5′-GAAGATGGTGATGGGATTTC-3′
28-2.2	F	5′-TAGACCGTCGTGAGACAGGT-3′
	R	5′-ATTGGCTCCTCAGCCAAGCA-3′
18-1702	F	5′-TCCCTGCCCTTTGTACACA-3′
	R	5′-GGCCGATCCGAGGGCCTCA-3′
28-2.1 (Internal control)	F	5′-TAGACCGTCGTGAGACAGGT-3′
	R	5′-CAACACATCATCAGTAGGGT-3′
Ph-PA136	F	5′-TTGATCGAGCCAGTAAAGGC-3′
	R	5′-AATCACATCCTTGGCGAACG-3′

## Data Availability

The original contributions presented in this study are included in the article and Appendix A. Further inquiries can be directed to the corresponding author.

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
