# Peer review of "Visible-Light-Driven Degradation of Biological Contaminants on the Surface of Textile Fabric Modified with TiO2-N Photocatalyst"

_ijms, 2025, doi:10.3390/ijms26157550_

Round 1
Reviewer 1 Report
Comments and Suggestions for Authors
The manuscript mainly reported that surface of blended cotton/polyester fabric was functionalized with N-doped TiO2 nanoparticles and applied for the oxidation of acetone vapor, degradation of DNA fragments of various lengths, and inactivation of PA136 virus bacteriophage and Candida albicans fungi. However, there are some obvious issues in this manuscript, which are listed as follows:
- No more than five keywords was suggested.
- It needs to be clarified where nitrogen comes from in N-doped TiO2. Will different nitrogen sources have different effects?
- EDS mapping is suggested to be added to illustrate the dispersion of copper atoms in PF-TN-Cu materials. Meanwhile, the corresponding TEM also should be added.
- N 1s and Cu 1s XPS spectrum of PF-TN-Cu should be added, especially for the valence state of copper.
- The vertical axis of UV–VIS diffuse reflectance spectra needs to be marked.
- Is copper or copper species present on the XRD spectrum? What is the load capacity of copper?
- How about the photocatalytic activity of TN-Cu? Compared with pure TN, the photocatalytic activity of PF-TN-Cu is worse. This seems to contradict the meaning expressed in this article.
- It is recommended to supplement the adsorption performance test of the photocatalyst for pollutants before determining its photocatalytic performance.
- How stable is the catalyst? Is there a significant difference in catalyst structure before and after the reaction. Relevant characterization should be supplemented.
Author Response
Response to the Reviewer #1:
Comment #1. No more than five keywords was suggested.
Response: Thank you for your comment. The Guide for Authors of IJMS recommends including three to ten pertinent keywords. We reduced the number of keywords to five according to your recommendation. Please, find the revised list as follows:
self-cleaning textile; photoactive fabric; photocatalytic degradation; N-doped TiO2; visible light
Comment #2. It needs to be clarified where nitrogen comes from in N-doped TiO2. Will different nitrogen sources have different effects?
Response: Thank you for your comment. In this study, visible-light active N-doped TiO2 (TiO2-N) was synthesized via our developed method using TiOSO4 precursor and ammonia as a precipitating agent, as well as a source of nitrogen. A pH-controlled precipitation of TiO2 from an aqueous solution of TiOSO4 using ammonia followed by the calcination of precipitate in air at an increased temperature results in the formation of nanocrystalline anatase, which can absorb radiation in the region up to 540 nm due to the presence of nitrogen impurities in the TiO2 lattice. Thus, you are right that nitrogen comes from ammonia. Experimental evidence and deep discussion on nitrogen state and its effect on light absorbance and photoactivity of the as-prepared TiO2-N can be found in our previous publications [R1,R2,R3]. Various nitrogen sources (i.e., urea, ammonium hydroxide, ammonium nitrate, diethanolamine, and triethylamine) are known to be used for the synthesis of visible-light active N-doped TiO2, and these nitrogen precursors provide slightly different effects on the UV-Vis diffuse reflectance spectrum of formed TiO2-N and its photocatalytic properties. Our results show that N-doped TiO2 prepared from TiOSO4 and ammonia precursors according to the proposed technique exhibits great visible-light activity in the degradation of organic pollutants, especially under blue. For this reason, it was employed for the preparation of self-cleaning textile fabric able to decompose biological contaminants under visible light.
[R1] N. Kovalevskiy, D. Svintsitskiy, S. Cherepanova, S. Yakushkin, O. Martyanov, S. Selishcheva, E. Gribov, D. Kozlov, D. Selishchev, Visible-Light-Active N-Doped TiO2 Photocatalysts: Synthesis from TiOSO4, Characterization, and Enhancement of Stability Via Surface Modification, Nanomaterials 12 (2022) 4146. https://doi.org/10.3390/nano12234146.
[R2] M. Lyulyukin, N. Kovalevskiy, A. Bukhtiyarov, D. Kozlov, D. Selishchev, Kinetic Aspects of Benzene Degradation over TiO2-N and Composite Fe/Bi2WO6/TiO2-N Photocatalysts under Irradiation with Visible Light, International Journal of Molecular Sciences 24 (2023) 5693. https://doi.org/10.3390/ijms24065693
[R3] E. Gribov, E. Koshevoy, T. Fazliev, M. Lyulyukin, D. Kozlov, D. Selishchev, Effect of surface Fe- and Cu-species on the flat-band potential and photoelectrocatalytic properties of N-doped TiO2, Journal of Photochemistry and Photobiology A: Chemistry 464 (2025) 116342. https://doi.org/10.1016/j.jphotochem.2025.116342.
We have mentioned the source of nitrogen in the main text of the manuscript. The revised version now reads as follows:
Visible-light-active TiO₂-N, known for its ability for complete mineralization of various pollutants [50], was employed as the photocatalytic component to obtain photoactive fabric with self-cleaning properties. TiO₂-N was prepared via the pH-controlled precipitation of titania from an aqueous solution of TiOSO4 using ammonia as a precipitating agent, as well as a source of nitrogen, followed by the calcination of precipitate in air at an increased temperature. This method resulted in the formation of nanocrystalline anatase, which can absorb radiation in the region up to 540 nm due to the incorporation of nitrogen impurities from ammonia into the TiO2 lattice.
Comment #3. EDS mapping is suggested to be added to illustrate the dispersion of copper atoms in PF-TN-Cu materials. Meanwhile, the corresponding TEM also should be added.
Response: Thank you for your comment. According to your recommendation, we have added element mapping for PF-TN-Cumaterial to illustrate Cu distribution. Concerning TEM, textile substrates are commonly not stable under exposure to high-power electron beam used in conventional TEM microscopes. Therefore, we have employed scanning transmission electron microscopy (STEM) to study the material. Please find the revised version of Figure 1 below:
Figure 1. SEM micrographs of (a) initial fabric (IF) and prepared photoactive (b,c) PF-TN and (d) PF-TN-Cu materials; (e) bright-field STEM image of coated fiber in PF-TN-Cu material; (f) copper distribution in PF-TN-Cu material according to EDX analysis.
Comment #4. N 1s and Cu 1s XPS spectrum of PF-TN-Cu should be added, especially for the valence state of copper.
Response: Thank you for your comment.
In the case of nitrogen, the synthesized N-doped TiO2 powder contained ~0.4 wt % of nitrogen in its composition [R2] that was high enough to be detected via the XPS method. Cu-modified TiO2-N (TN-Cu) with Cu loading of 1 wt.% also exhibited an XPS signal with intensity high enough to be detected in Cu-related regions and recognized. Both corresponding spectra and their deconvolution are shown in Figure S1a,b in the Supporting Information.
However, no signals in the N 1s and Cu 2p regions were observed in the XPS spectra of photoactive fabrics (Figure S1c,d). This result is due to extremely low content of these elements in the composition of the PF-TN and PF-TN-Cu materials. Nitrogen and copper contents in PF-TN-Cu are estimated as 0.004 and 0.01 wt.%, respectively, and these values are much lower than the detection limit of the XPS method.
A brief discussion of these data has been added to the manuscript. Please find the revised version below.
[R2] M. Lyulyukin, N. Kovalevskiy, A. Bukhtiyarov, D. Kozlov, D. Selishchev, Kinetic Aspects of Benzene Degradation over TiO2-N and Composite Fe/Bi2WO6/TiO2-N Photocatalysts under Irradiation with Visible Light, International Journal of Molecular Sciences 24 (2023) 5693. https://doi.org/10.3390/ijms24065693.
Supporting Information reads as follows:
- XPS spectra of TN-CU, PF-TN, and PF-TN-Cu samples in (a) N 1s and (b) Cu 2p regions
Figure S1. XPS spectra of Cu-modified TiO2-N (TN-Cu) powder with Cu loading of 1 wt.% in (a) N 1s and (b) Cu 2p spectral regions; XPS spectra of photoactive fabric modified with TiO2-N (PF-TN) and Cu-modified photoactive fabric with TiO2-N (PF-TN-Cu) in (c) N 1s and (d) Cu 2p spectral regions.
The revised manuscript reads as follows:
It is worth noting that no XPS signals are observed in the photoelectron N 1s and Cu 2p spectral regions of PF-TN-Cu material, although nitrogen and copper states are evident for the powdered Cu-modified TiO2-N photocatalyst (TN-Cu) by the means of XPS method (see Figure S1 in the Supporting Information). This fact results from extremely low content of these elements due to the predominance of fabric and TiO2 components in the composition of photoactive material. Nitrogen and copper contents in PF-TN-Cu are estimated as 0.01 and 0.03 wt.%, respectively, and these values are much lower than the detection limit of the XPS method.
Comment #5. The vertical axis of UV–VIS diffuse reflectance spectra needs to be marked.
Response: Thank you for your valuable suggestion. We have marked the vertical axis of the UV–VIS diffuse reflectance spectra in the main text of the manuscript. Please find the revised version of this figure below:
Figure 2. (a) XPS spectra in the Ti2p region, (b) XRD patterns, and (c) UV–VIS diffuse reflectance spectra of the initial fabric (IF), photoactive fabric modified with TiO2-N (PF-TN), and photoactive fabric modified with TiO2-N and Cu species (PF-TN-Cu).
Comment #6. Is copper or copper species present on the XRD spectrum? What is the load capacity of copper?
Response: Thank you for your comment. No peaks corresponding to copper or copper-containing species were observed in the XRD patterns of the photoactive PF-TN-Cu fabric. This result is attributed to extremely low Cu content (0.03 wt%) in the composition of PF-TN-Cu material. The concentration of the copper precursor (Cu(OAc)₂) used during the preparation was 0.28 g L⁻¹, as shown in Table 1 of the manuscript. We have added this information regarding the copper loading capacity to the main text. The revised version reads as follows:
The copper-modified material (PF-TN-Cu) was synthesized by addition of copper(II) acetate monohydrate to the impregnated composition described above. The amount of added Cu(OAc)2 was estimated on the basis to achieve Cu loading of 1 wt% toward the mass of TiO2-N component (Table 1) because it was regarded as an optimum value [58]. The other parameters of the synthesis procedure were kept the same. Table 1 shows the composition of each sample, and the quantities of the components used.
Comment #7. How about the photocatalytic activity of TN-Cu? Compared with pure TN, the photocatalytic activity of PF-TN-Cu is worse. This seems to contradict the meaning expressed in this article.
Response: Thank you for your valuable comment. There are no contradictions in the results discussed in this study:
The first point is that the photocatalytic activity of photoactive textile materials is much lower than the activity of corresponding powdered photocatalysts due to low content of photoactive component and, consequently, much thinner photoactive layer compared to a thick layer (30 mg cm−2 in our tests) of powdered photocatalysts that means lower amount of light absorbed and lower reaction rate. Therefore, the activity of textile PF-TN material is much lower than the activity of a thick layer of TN powder (see Figure 3 in the main text of manuscript).
The photocatalytic activity of the powdered TN-Cu photocatalyst (1 wt.% of Cu) under visible light is ca. 30% higher than the activity of N-doped TiO2 (TN) powder (see Figure S2a in the Supporting Information and [R1]) because of copper species, which are distributed on photocatalyst’s surface [R3], improve the separation of photogenerated charges, thus increasing the activity of the photocatalyst. However, when comparing textile PF-TN and PF-TN-Cu materials, their photocatalytic activity is similar in both spectral regions (see Figure 3 in the main text of manuscript). Probably, the concentration of Cu species on the surface of TiO2-N crystallites in the case PF-TN-Cu is lower than in the case of powdered TN-Cu photocatalyst because many Cu species are absorbed on the surface of fabric fibers and amorphous titania binder. As a result, the surface concentration of Cu species on the surface TiO2-N crystallites is not high enough to substantially improve the separation of charges photogenerated in TiO2-N crystallites and enhance the overall photocatalytic activity of the material. Optimization of Cu content will be performed in our next studies.
[R1] N. Kovalevskiy, D. Svintsitskiy, S. Cherepanаova, S. Yakushkin, O. Martyanov, S. Selishcheva, E. Gribov, D. Kozlov, D. Selishchev, Visible-Light-Active N-Doped TiO2 Photocatalysts: Synthesis from TiOSO4, Characterization, and Enhancement of Stability Via Surface Modification, Nanomaterials 12 (2022) 4146. https://doi.org/10.3390/nano12234146
[R3] E. Gribov, E. Koshevoy, T. Fazliev, M. Lyulyukin, D. Kozlov, D. Selishchev, Effect of surface Fe- and Cu-species on the flat-band potential and photoelectrocatalytic properties of N-doped TiO2, Journal of Photochemistry and Photobiology A: Chemistry 464 (2025) 116342. https://doi.org/10.1016/j.jphotochem.2025.116342
The revised manuscript reads as follows:
Regardless the fact that the visible-light activity of Cu-modified TiO2-N (TN-Cu) photocatalyst with Cu loading of 1 wt% is ~30% higher than the activity of initial TiO2-N (TN) sample (Figure S2a in the Supporting Information), the activity of PF-TN-Cumaterial similarly equals to 0.1 μmol min–1 under visible light as for PF-TN (Figure 3). This means no noticeable effect of Cu species on the material activity under visible light. The activity of TiO2-N powder is known to increase after copper modification due to improving the transfer of photogenerated holes to the photocatalyst surface [58]. Probably, the concentration of Cu species on the surface of TiO2-N crystallites in the case PF-TN-Cu is lower than in the case of powdered TN-Cu photocatalyst because many Cu species are absorbed on the surface of fabric fibers and amorphous titania binder. As a result, the surface concentration of Cu species on the surface TiO2-N crystallites is not high enough to substantially improve the separation of charges photogenerated in TiO2-N crystallites and enhance the overall photocatalytic activity of the material. An additional argument in favor of copper adsorption on fabric fibers is the evidence of improved adsorption of copper ions in the presence of hydroxyl groups on fibers [62,63]. Under UVA radiation, PF-TN-Cu exhibits a slight decrease in the activity compared to PF-TN, but the observed activity is still high enough for productive oxidation of chemicals.
Comment #8. It is recommended to supplement the adsorption performance test of the photocatalyst for pollutants before determining its photocatalytic performance.
Response: Thank you for your comment. The main goal for creating photoactive self-cleaning fabrics is the complete degradation of pollutants on their surface. Therefore, this paper focuses on the degradation of different types of biological pollutants over photoactive self-cleaning fabrics under their exposure to light. For illustration of the adsorption properties of studied materials, the paper presents the results of experiments on the surface of initial unmodified fabric (IF) and photoactive material (PF-TN) without light exposure. The results of experiments show that adsorption has minor effect compared to the photocatalytic degradation. The deposition of photoactive coating enhances initial adsorption of the pollutants; however, most of concerned biological objects remain stable on the surface of PF-TN without exposure to light.
In the case of test reaction of acetone vapor oxidation, these experiments were carried out in a continuous-flow reactor, and the photocatalytic activity was evaluated after complete achievement of adsorption-desorption equilibrium.
Comment #9. How stable is the catalyst? Is there a significant difference in catalyst structure before and after the reaction. Relevant characterization should be supplemented.
Response: Thank you for your comment. According to your recommendation, we have added to the Supporting Information the data that illustrate high stability of TiO2-N photocatalyst modified with copper (1 wt.%). Figure S2b shows that TN-Cu photocatalyst exhibits steady-state activity in the test reaction of acetone vapor oxidation after irradiation more than 60 h. There were not any changes in the crystal structure of photocatalyst after this stability test. Our previous study [R4] also showed that similar TiO2-coated photoactive fabric had a high level of activity during a long time.
It is worth noting that the durability of materials after repeated washings is more important for functionalized fabrics because laundry treatment can lead to washing out the photoactive coating. To illustrate this aspect, we have added to the Supporting Information a section concerning the stability of photocatalytic activity of photoactive material after several cycles in a washing machine. A brief discussion of these results was added to the main text of the manuscript.
[R4] Solovyeva, M.; Selishchev, D.; Cherepanova, S.; Stepanov, G.; Zhuravlev, E.; Richter, V.; Kozlov, D. Self-Cleaning Photoactive Cotton Fabric Modified with Nanocrystalline TiO2 for Efficient Degradation of Volatile Organic Compounds and DNA Contaminants, Chemical Engineering Journal 388 (2020) 124167. doi:10.1016/j.cej.2020.124167.
The revised manuscript reads as follows:
- 6: Note that application of the titanium(IV) isopropoxide binder in the impregnating dispersion provides incorporation of photocatalyst’s nanoparticles into the titania matrix and their attachment to the surface of the fabric fibers. As a result, the as-prepared materials have a good stability toward washing procedures (Figure S3). Another aspect shown in our previous paper [49] is that the TiO2-coated fabrics exhibit steady-state photocatalytic activity under long-term irradiation for days. All these factors confirm their potential for practical application as self-cleaning textile fabrics.
Supporting Information reads as follows:
- Photocatalytic activity of powdered TN-Cu photocatalyst
Figure S2. (a) Comparison of initial photocatalytic activity of TiO2-N (TN) powder and Cu-modified TiO2-N (TN-Cu) powder with Cu loading of 1 wt.% in the test reaction of acetone vapor oxidation under visible light; (b) stability of TN-Cu photocatalyst in the oxidation of acetone vapor under long-term irradiation.
- Durability of photoactive PF-TN-Cu fabric after repeated washing
The stability of photoactive coating was studied in several laundry cycles in a washing machine by measuring its photocatalytic activity in the test reaction of acetone vapor oxidation under UVA light after each washing cycle. Briefly, a piece of photoactive PF-TN-Cu material was placed in a washing machine (Indesit, Belarus) and washed according to a standard 30-min program proposed by the manufacturer for daily washing using a detergent. Washed piece of fabric was dried at 70 °С for 12 h, and its photocatalytic activity was then evaluated according to the technique described in the main text of manuscript. The procedures of washing, drying, and activity measurement were repeated three times more using the same piece of fabric to monitor a change in the activity. Figure S3 shows that the activity of PF-TN-Cu material strongly decreases from 0.29 to 0.22 mmol min–1 after the first cycle of washing, probably due to washing out the most weakly bonded particles, but then it decreases slightly and keeps more than 60% (0.175 mmol min–1) initial activity. Then, two additional long-term (1.5–2 h) washing procedures were performed to intensify laundry impact. This long-term treatment leads to a decrease in activity down to 0.13 mmol min–1 that corresponds to 40% of initial level but it remains at the same level as the number of washing cycles increases. This result indicates the stability of the prepared photoactivity fabric toward washing procedures.

Reviewer 2 Report
Comments and Suggestions for Authors
The article is a high-quality and relevant study demonstrating the potential of using TiO₂-N and copper to create self-cleaning textile materials. The results have significant potential for practical application, especially in the medical field. Despite minor comments, the work deserves high marks and can be recommended for publication after taking into account the above comments.
- Although the authors note that the coating exhibits good stability, it would be useful to include data on the durability of the material after repeated washing or mechanical stress.
- The potential impact of the modified fabric on human skin, especially with prolonged contact, is not discussed. This is important for medical applications.
- The paper could benefit from a more detailed comparison with other photocatalytic materials, such as Ag or ZnO based, to highlight the advantages of TiO₂-N.
- Experimental conditions (e.g. light intensity, humidity) could have been more detailed to facilitate reproducibility of results.
Author Response
Response to the Reviewer #2:
Comment #1. Although the authors note that the coating exhibits good stability, it would be useful to include data on the durability of the material after repeated washing or mechanical stress.
Response: Thank you for your valuable suggestion. We have added to the Supporting Information a section concerning the stability of photocatalytic activity of photoactive material after several laundry cycles in a washing machine. A brief discussion of these results was added to the main text of the manuscript.
The revised manuscript reads as follows:
Note that application of the titanium(IV) isopropoxide binder in the impregnating dispersion provides incorporation of photocatalyst’s nanoparticles into the titania matrix and their attachment to the surface of the fabric fibers. As a result, the as-prepared materials have a good stability toward washing procedures (Figure S3). Another aspect shown in our previous paper [49] is that the TiO2-coated fabrics exhibit steady-state photocatalytic activity under long-term irradiation for days. All these factors confirm their potential for practical application as self-cleaning textile fabrics.
Supporting Information reads as follows:
- Durability of photoactive PF-TN-Cu fabric after repeated washing
The stability of photoactive coating was studied in several laundry cycles in a washing machine by measuring its photocatalytic activity in the test reaction of acetone vapor oxidation under UVA light after each washing cycle. Briefly, a piece of photoactive PF-TN-Cu material was placed in a washing machine (Indesit, Belarus) and washed according to a standard 30-min program proposed by the manufacturer for daily washing using a detergent. Washed piece of fabric was dried at 70 °С for 12 h, and its photocatalytic activity was then evaluated according to the technique described in the main text of manuscript. The procedures of washing, drying, and activity measurement were repeated three times more using the same piece of fabric to monitor a change in the activity. Figure S3 shows that the activity of PF-TN-Cu material strongly decreases from 0.29 to 0.22 mmol min–1 after the first cycle of washing, probably due to washing out the most weakly bonded particles, but then it decreases slightly and keeps more than 60% (0.175 mmol min–1) initial activity. Then, two additional long-term (1.5–2 h) washing procedures were performed to intensify laundry impact. This long-term treatment leads to a decrease in activity down to 0.13 mmol min–1 that corresponds to 40% of initial level but it remains at the same level as the number of washing cycles increases. This result indicates the stability of the prepared photoactivity fabric toward washing procedures.
Figure S3. Durability of photoactive fabric after several washing cycles.
Comment #2. The potential impact of the modified fabric on human skin, especially with prolonged contact, is not discussed. This is important for medical applications
Response: Thank you for your valuable comment. We have added the information on material’s biosafety in the main text of the manuscript. The revised manuscript now reads as follows:
The obtained photoactive fabrics functionalized with TiO2-N can provide degradation of various biological contaminants on their surface under visible light or UVA radiation. The self-cleaning ability of materials under visible light shows promise for their usage in creating fabric products, such as medical coats and curtains, for medical and scientific institutions that require high levels of environmental cleanliness. The possibility of their practical application in the mentioned areas is also provided by the low cytotoxicity of used functional components toward human skin cells, even with prolonged contact [37,97]. Although photoactive fabrics functionalized with nanocrys-talline photocatalyst have a potential risk of leaching photocatalyst’s nanoparticles [98], the proposed preparation technique solves this problem via using of the titanium(IV) isopropoxide binder, which forms the titania matrix and incorporates photocatalyst’s particles. Thus, they do not represent as single nanoobjects, and the as-functionalized photoactive fabrics exhibit low environmental impact.
[37] Stan, M.S.; Badea, M.A.; Pircalabioru, G.G.; Chifiriuc, M.C.; Diamandescu, L.; Dumitrescu, I.; Trica, B.; Lambert, C.; Dinischiotu, A. Designing Cotton Fibers Impregnated with Photocatalytic Graphene Oxide/Fe, N-Doped TiO2 Particles as Prospective Industrial Self-Cleaning and Biocompatible Textiles. Materials Science and Engineering: C 2019, 94, 318–332, doi:10.1016/j.msec.2018.09.046.
[97] Nica, I.C.; Stan, M.S.; Dinischiotu, A.; Popa, M.; Chifiriuc, M.C.; Lazar, V.; Pircalabioru, G.G.; Bezirtzoglou, E.; Iordache, O.G.; Varzaru, E.; et al. Innovative Self-Cleaning and Biocompatible Polyester Textiles Nano-Decorated with Fe–N-Doped Titanium Dioxide. Nanomaterials 2016, 6, 214, doi:10.3390/nano6110214.
[98] Busi, E.; Maranghi, S.; Corsi, L.; Basosi, R. Environmental Sustainability Evaluation of Innovative Self-Cleaning Textiles. Journal of Cleaner Production 2016, 133, doi:10.1016/j.jclepro.2016.05.072.
Comment #3. The paper could benefit from a more detailed comparison with other photocatalytic materials, such as Ag or ZnO based, to highlight the advantages of TiO₂-N.
Response: Thank you for your valuable suggestion. Information on the modification of fabric materials with Ag and ZnO was added to the main text of the manuscript. The revised manuscript now reads as follows:
Among these, self-cleaning textiles warrant special attention, as fabrics can harbor pathogenic microorganisms that remain infectious for long-term periods [9]. Antibacterial properties can be achieved via the decoration of fabric’s surface with silver and copper nanoparticles [10,11]. The functionalization of fabrics with photocatalysts, such as titanium dioxide (TiO2) and zinc oxide (ZnO), is another promising approach to obtain self-cleaning textiles with a high oxidation activity in chemical degradation and micro-organism disinfection [12].
[10] Xing, H.; Cheng, J.; Tan, X.; Zhou, C.; Fang, L.; Lin, J. Ag Nanoparticles-Coated Cotton Fabric for Durable Antibacterial Activity: Derived from Phytic Acid–Ag Complex. The Journal of The Textile Institute 2020, 111, 855–861.
[11] Xu, Q.; Ke, X.; Ge, N.; Shen, L.; Zhang, Y.; Fu, F.; Liu, X. Preparation of Copper Nanoparticles Coated Cotton Fabrics with Durable Antibacterial Properties. Fibers Polym 2018, 19, 1004–1013, doi:10.1007/s12221-018-8067-5.
As TiO2 has a bandgap of 3.0–3.2 eV [18], the TiO2-coated fabrics can be activated under UV radiation only. ZnO is also commonly used as the photoactive component for the modification of textile materials to provide them self-cleaning properties. Similarly to TiO2, the ZnO-coated fabrics can decompose contaminants under the radiation of UV region due to a wide bandgap of ZnO (3.2–3.4 eV) [19]. This fact limits the efficiency of solar light utilization because the UV region is less than 5% of the solar spectrum [20,21].
[19] Bogdan, J.; Zarzyńska, J.; Pławińska-Czarnak, J. Comparison of Infectious Agents Susceptibility to Photocatalytic Effects of Nanosized Titanium and Zinc Oxides: A Practical Approach. Nanoscale Research Letters 2015, 10, 309, doi:10.1186/s11671-015-1023-z.
Comment #4. Experimental conditions (e.g. light intensity, humidity) could have been more detailed to facilitate reproducibility of results.
Response: Thank you for your valuable suggestion. As the samples were placed directly under the light source at a distance, we measured light intensity at this distance and mentioned it in the text as the irradiance (mW cm-2). The values of measured irradiance were mentioned in the manuscript next to the wavelength of the LED used. In the revised version, we have specified this fact. Also, we have added the information on distance from the sample to the LED, as well as air humidity and temperature for the experiments with biological contaminants for more detailed description to facilitate reproducibility of results into the manuscript. Please find the revised version below.
Briefly, each fabric sample (9 cm2) was placed into the continuous-flow photoreactor and irradiated sequentially with two light-emitting diodes (LEDs) provided blue light (λmax = 450 nm) or UVA light (λmax = 365 nm). In each case, the sample was located directly under the LED at a distance of 9 cm, and the measured irradiance of sample was 160 mW cm–2 and 10 mW cm–2 for blue light and UVA light, respectively… Volume flow rate of humidified air (relative humidity of 20%) was 0.065 L min−1…
Degradation of biological contaminates on the surface of fabric materials was studied using the fragments of DNA molecules of various lengths, PA136 virus bacteriophage, and Candida albicans fungi under exposure to radiation from a low-pressure mercury lamp (Aerolife LLC, Moscow, Russia) provided UVA light (λmax = 365 nm, 3 mW cm–2) or two LEDs provided UVA light (λmax = 370 nm, 5 mW cm–2) and blue light (λmax = 450 nm, 30–80 mW cm–2), respectively. The samples were located directly under the light source at a distance of 11–15 cm, and the actual values of measured irradiance are mentioned in the corresponding sections below. All experiments were carried out at a room temperature of 23–25 °С and relative humidity of 20%simultaneously using photoactive fabric materials and the initial fabric as a control…
